# Minimally Invasive L5 Corpectomy with Navigated Expandable Vertebral Cage: A Technical Note

**DOI:** 10.3390/brainsci11091241

**Published:** 2021-09-19

**Authors:** Taro Yamauchi, Ashish Jaiswal, Masato Tanaka, Yoshihiro Fujiwara, Yoshiaki Oda, Shinya Arataki, Haruo Misawa

**Affiliations:** 1Department of Orthopaedic Surgery, Okayama Rosai Hospital, Okayama 702-8055, Japan; ygitaro0307@yahoo.co.jp (T.Y.); drashishjaiswal@yahoo.com (A.J.); fujiwarayoshihiro2004@yahoo.co.jp (Y.F.); odaaaaaaamn@yahoo.co.jp (Y.O.); araoyc@gmail.com (S.A.); 2Department of Orthopaedic Surgery, Okayama University Hospital, Okayama 700-0914, Japan; misharu@md.okayama-u.ac.jp

**Keywords:** L5 corpectomy, minimally invasive surgery, navigation, single lateral position, C-arm free

## Abstract

Background: Conventional L5 corpectomy requires a large incision and an extended period of intraoperative fluoroscopy. We describe herein a new L5 corpectomy technique. Methods: A 79-year-old woman was referred to our hospital for leg pain and lower back pain due to an L5 vertebral fracture. Her daily life had been affected by severe lower back pain and sciatica for more than 2 months. We initially performed simple decompression surgery, but this proved effective for only 10 months. Results: For revision surgery, the patient underwent minimally invasive L5 corpectomy with a navigated expandable cage without fluoroscopy. The second surgery took 215 min, and estimated blood loss was 750 mL. The revision surgery proved successful, and the patient could then walk using a cane. In terms of clinical outcomes, the Oswestry Disability Index improved from 66% to 24%, and the visual analog scale score for lower back pain improved from 84 to 31 mm at the 1-year follow-up. Conclusions: Minimally invasive L5 corpectomy with a navigated expandable vertebral cage is effective for reducing cage misplacement and surgical invasiveness. With this new technique, surgeons and operating room staff can avoid the risk of adverse events due to intraoperative radiation exposure.

## 1. Introduction

Thoracolumbar junction shows several features compared with other spine areas. Surgical approaches to pathologies involving the L5 vertebra constitute a significant challenge due to its anatomical features [1]. The biomechanics of this area are characterized by a high shearing force and compressive forces [2]. The vascular anatomy of this area is characterized by the locations of the great vessels. As a result, L5 corpectomy has been considered to have a high complication rate [3]. The problem with major surgeries such as corpectomy and long posterior corrective fusion in elderly patients is the high rate of complications [4]. Therefore, minimally invasive surgery (MIS) has been receiving increasing attention as a method of reducing morbidity and perioperative complications for spinal surgery [5]. A key disadvantage of conventional MIS corpectomy is the misplacement of vertebral cages and the need for extended use of intraoperative fluoroscopy [6]. Computer-assisted spinal surgery is the current trend for spinal surgeries, and three-dimensional image guidance technology is available for spinal MIS [7]. Another concern for MIS surgeons is radiation exposure when performing MIS under C-arm guidance [4]. Considering the above issues, we report a new technique for MIS L5 corpectomy with a navigated expandable cage under navigation guidance.

## 2. Case Presentation

The institutional ethics committee approved this study (approval no. 305). The patient provided written informed consent.

### 2.1. Patient History

A 79-year-old woman was referred to our orthopedic department with pain in the leg and lower back. Over the previous 3 months, she had gradually become unable to walk, so she had visited another hospital and received conservative treatment. Activities of daily living were severely affected by severe lower back pain and numbness and weakness of the left leg for more than 2 months.

### 2.2. Physical Examination

The patient could not walk or stand unaided. No hyporeflexia of the legs was detected on examination, and she described severe lower back pain with a limited range of spinal motion. Numbness was identified on both legs at the L5 and S1 dermatomes. Power in both lower legs was grade 4. No urinary or bowel dysfunction was evident.

### 2.3. Preoperative Imaging

Radiography at the initial visit demonstrated L5 vertebral collapse with slight instability. Preoperative computed tomography (CT) showed a collapsed bony fragment protruding into the spinal canal. Preoperative magnetic resonance imaging (MRI) revealed severe spinal canal stenosis at the L4–5 and L5–S1 levels (Figure 1A,B). The patient showed severe osteoporosis, with a bone mineral density of 0.856 g/cm^2^ (−1.8 SD).

### 2.4. First Surgery and Postoperative Images

Decompression surgery was initially performed, considering the slight instability and the age of the patient. Postoperative images indicated good decompression of the spinal canal (Figure 1C,D). She received moderate pain relief and could walk more than 300 m after surgery.

### 2.5. Second Surgery and Postoperative Images

After 9 months, symptoms had again worsened, and she revisited our hospital. Follow-up images showed that the L5 vertebra had refractured, increasing lumbar instability at this level. CT demonstrated a pincer-type fracture and MRI indicated dural compression (Figure 2A,B). Fusion MR-CT images revealed an abnormal course of the left common iliac artery (Figure 2C,D). As a result of the severity of symptoms, we decided to perform L5 MIS corpectomy with a navigated expandable cage, using a posterior percutaneous pedicle screw for the fixation of L3 to the pelvis. The second surgery took 215 min, with an estimated blood loss of 750 milliliters. No intra- or postoperative complications were encountered. Postoperative images showed good spinal canal decompression and restoration of spinal alignment (Figure 3A,B). Postoperative CT showed satisfactory cage position (Figure 3C,D).

### 2.6. Follow-Up Results

After 4 months, the patient was capable of almost normal activity. Muscle weakness had resolved in both legs (power grade 5/5). At the 1-year follow-up, CT demonstrated a slight L4 fracture, but solid bony fusion from L3 to the pelvis and maintenance of good alignment were also seen (Figure 4). Clinical outcomes had improved at the 1-year follow-up. The Oswestry Disability Index improved from 66% to 24%, and lower back pain, as assessed using a visual analog scale, improved from 84 to 31 mm.

## 3. Operative Procedure

### 3.1. Mini-Extraperitoneal Approach

The patient is placed in the right lateral decubitus position on an adjustable hinged operating carbon table. The percutaneous reference frame for navigation is fixed at the left sacroiliac joint (Figure 5). Then, the O-arm is positioned, and three-dimensional reconstructed images are obtained and transmitted to the StealthStation surgical navigation system Spine 7 ^R^. Navigated spinal instruments are registered, and a left oblique 5 cm skin incision is made along the best point, which is marked with a navigation pointer (Figure 6A,B). The tip of the left iliac wing should be removed for a vertical approach to the L5 vertebra. The retroperitoneal space is separated by blunt finger dissection, and the psoas major muscle, vessels, and L5 vertebra are exposed. After the first navigated probe is safely positioned on the psoas major muscle, sequential dilation is used until 22 mm is reached. The special self-retaining retractor is placed in the correct position. If necessary, an L3–4 oblique lumbar interbody fusion cage can be inserted to enhance cranial screw anchors (Figure 6C,D).

### 3.2. Corpectomy with Navigation

First, the left ureter and left common iliac vessels are identified. The most important point is the ascending lumbar vein, and the iliolumbar vein should be identified, ligated, or clipped, and then cut. The discs of L4–5 and L5–S1 are exposed, and thorough discectomy is performed using Kerrison rongeurs, pituitary forceps, navigated shavers, a navigated Cobb elevator, and navigated ring curettes. The navigated osteotome is used to remove the L5 vertebra (Figure 7A,D). In the same single position, percutaneous pedicle screws are inserted simultaneously during neuromonitoring (Figure 7B,C).

### 3.3. Navigated Expandable Vertebral Cage

After complete resection of the collapsed vertebra, the required cage size is measured with a navigated trial (Figure 8A,B). When the trial is completed, a mixture of iliac bone and demineralized bone matrix are inserted into the cage. Then, a special expandable vertebral cage, T2 Stratosphere^TM^ Expandable Corpectomy System, is inserted during navigation guidance (Figure 8C,D). If necessary, intraoperative fluoroscopy is recommended to expand the cage, because the navigation monitor cannot display real-time expansion. For patients with severe osteoporosis, posterior long fusion (two levels above, sacroalar iliac fixation) is recommended for L5 corpectomy. An additional one level above oblique lumbar interbody fusion is recommended to secure the solid L3–4 fusion and prevent screw back-out.

## 4. Discussion

Osteoporotic vertebral fractures (OVF) are mostly treated conservatively, but surgical interventions are required in refractory cases and those with neurological deficits or deformity. When diagnosing OVF, MRI is clearly superior to both conventional radiography and CT and should be preferred as the first diagnostic examination [8]. OVF affect mostly elderly patients, so the percutaneous procedures of percutaneous vertebroplasty and balloon kyphoplasty have been the procedures of choice in the absence of neurological deficits and significant deformities [9,10,11]. However, aggressive surgical interventions are required in patients with neurological deficits and severe deformities, with the goal of neurological decompression and correction of the deformity [12]. Major operations in elderly populations have higher complication rates [1], so efforts to minimize surgical approaches without compromising the treatment objectives have been explored. Several options are available to reconstruct collapsed vertebrae, such as autogenous iliac crest bone graft, allograft, non-expandable cage [13], and expandable vertebral cage [4]. For severely osteoporotic patients, autogenous iliac bone graft is inadequate, because the bone quality is insufficient to maintain stability. Further, non-expandable cages are difficult to fit into the resected vertebral space. 

Our case demonstrated a novel minimally invasive simultaneous technique of L5 corpectomy with a navigated expandable cage and posterior percutaneous pedicle screw fixation using O-arm navigation. Surgical approaches to pathology of the L5 vertebra constitute a significant challenge due to the anatomical features of the lumbosacral junction [1]. The biomechanics of this area are characterized by high shearing force and compressive forces [2]. L5 corpectomy has a high complication rate, reportedly as high as 36% [2]. Furthermore, the presence of the great vessels in the retroperitoneal compartment compounds the surgical risk of vascular injury by as much as 13.8% [3]. L5 corpectomy from the anterior approach was first reported by Gaines in 1985, but this technique was applied only for high-grade spondylolisthesis cases [14]. L5 corpectomy is currently applied for burst fractures [2,15], infections [2,16], and tumors [15,17]. In our patient, surgery was performed in the lateral decubitus position to avoid the need to change position and drapes for anterior and posterior procedures, thereby reducing the operative time, risks of contamination, and inconvenience of re-registration for navigation purposes. Hiyama et al., reported an additional average repositioning time of 34 min between the lateral decubitus and prone positions [18]. Single-position surgery reduces the time and need for staffing associated with intraoperative repositioning and may provide significant cost savings [1]. Percutaneous posterior pedicle screw fixation in the lateral position per se can be technically challenging, but the use of navigation makes it feasible and accurate [4,19]. Another advantage of doing both procedures in the lateral decubitus position is that both concomitant procedures can be performed by two surgical teams simultaneously with the added benefit of diminished operative time and exhaustion. Prolonged surgical times in elderly patients with multiple serious comorbidities can be hazardous and at times, the two procedures may need to be done in a staged manner, leading to increased hospitalization and delayed mobilization, which itself has many risks [12]. Hence, single-position surgery and employment of two concomitant surgical teams may permit the performance of both procedures in a single stage. Furthermore, the lateral position tends to be better tolerated by the patient compared to prone surgery and avoids many possible concerns with prone positioning such as postoperative vision loss, cardiovascular complications, hypovolemia, reduced pulmonary compliance, and cardiac arrest [20].

The minimally invasive retroperitoneal approach for interbody fusion is well documented, but only sporadic reports of a similar approach for the more extensive procedure of corpectomy have been published [4,19,21]. Navigation makes corpectomy feasible through a small portal in particular, which is difficult to do at the L5 level as in the present case. The three main challenges in L5 corpectomy are the proximity of vascular structures rendering the anterior approach dangerous, the higher propensity of the transition zone for implant failure, and the unique anatomy with a high lordotic angle between the L4 endplate and sacrum making the fitting of straight cylindrical cages or allograft struts difficult [2]. To overcome these challenges, a lateral retroperitoneal approach with navigated expandable lordotic cage placement and augmentation with posterior pedicle screw instrumentation were applied in the present case. Accurate cage placement can be difficult between lordotic endplates after corpectomy [2]. Intraoperative fluoroscopy alone is unable to clearly show the correct three-dimensional position of the cage [4,21]. Yu et al. [21] reported the use of O-arm navigation for thoracolumbar corpectomy, but the cage in that case was not navigated. The new navigated expandable vertebral cage used in the present case has special features that facilitate insertion under navigation. The extended endcap has a self-adjusting mechanism with a total range of motion of 16°, allowing the endcap to fit the non-parallel gap and distribute surface contact evenly. Placing the cage in the correct position usually requires fluoroscopy. However, this cage is navigated, and so, it can be visualized in three dimensions in all planes on the navigation monitor [7]. 

In terms of utility, intraoperative navigation in spine surgery has stood the test of time, with many studies proving this method to be beneficial in terms of increased accuracy of pedicle screw placement [18,22]. The feasibility of wire-free percutaneous pedicle screw placement with navigated screws further avoids guidewire-related problems [4,7]. Anterolateral procedures such as osteotomies, interbody fusions, and corpectomies can be done more efficiently with the use of navigation and more so with the advent of navigated cages [4]. Particularly in obese patients, a navigation probe can be useful for selection of the incision site for minimally invasive procedure in line with the operative anterior spinal segment [4], thus avoiding intraoperative difficulties with misplaced incisions.

Radiation exposure to the surgeon and operating room staff is significantly reduced in terms of fluoroscopy time and exposure, from 168.7 s and 2.38 mSv per case to 32.7 s and 0.52 mSv per case in intraoperative CT-based navigation verses fluoroscopy-assisted corpectomy, respectively [23]. With our navigation technique, intraoperative fluoroscopy is unnecessary, so the radiation risk to the operating staff is zero.

Our new technique does show several disadvantages. First, intraoperative CT-based spine surgery is not without the limitations of possible surgical errors involving misplacement of spinal implants due to inadvertent movement of the reference frame. Second, establishment costs are higher. Third, because of the simultaneous technique, two surgeons are necessary to perform this technique.

## 5. Conclusions

Vertebroplasty or kyphoplasty for OVF is a standard technique. In severe cases such as paralysis, MIS L5 corpectomy with a navigated expandable vertebral cage is a safe and effective technique that reduces surgical time and intraoperative blood loss while maintaining sagittal and coronal corrections similar to conventional surgery. With this technique, accurate cage placement can be performed with navigation. This new procedure reduces radiation exposure to the surgeon and operating room staff compared with conventional fluoroscopic MIS techniques. 

## Figures and Tables

**Figure 1 brainsci-11-01241-f001:**
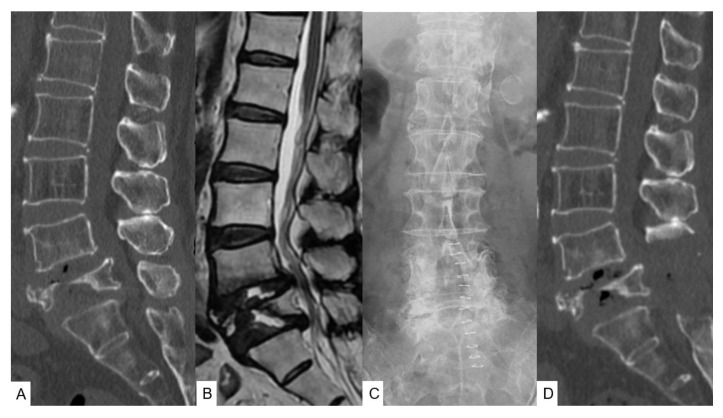
Pre- and postoperative images of the first surgery. (**A**) Preoperative mid-sagittal reconstruction CT; (**B**) Mid-sagittal MRI T2-weighted image; (**C**) Postoperative radiogram; (**D**) Postoperative mid-sagittal reconstruction CT.

**Figure 2 brainsci-11-01241-f002:**
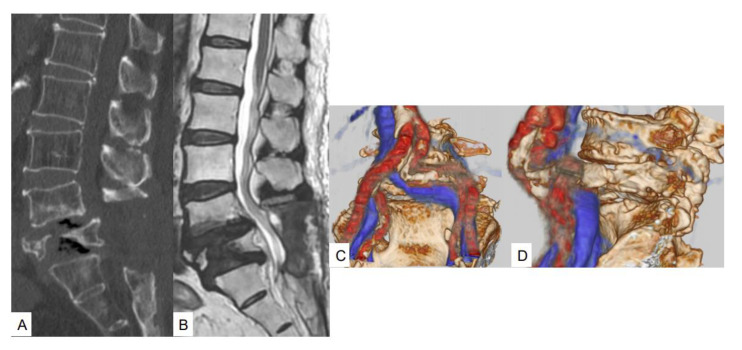
Images at the 9-month follow-up. (**A**) Mid-sagittal reconstruction CT; (**B**) Mid-sagittal T2-weighted image; (**C**) Anteroposterior 3D vascular image; (**D**) Lateral 3D vascular image.

**Figure 3 brainsci-11-01241-f003:**
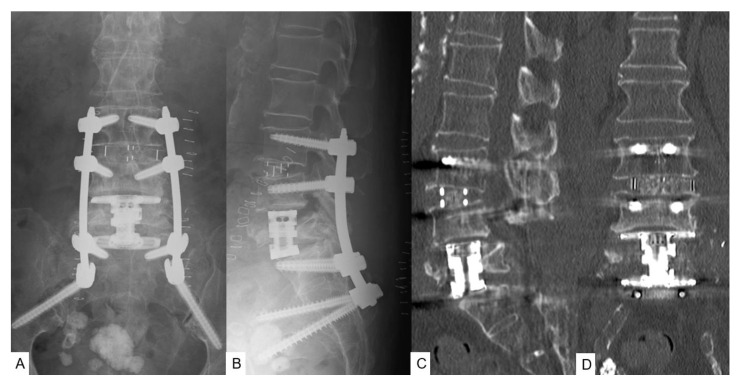
Postoperative images of second surgery. (**A**) Posteroanterior radiogram; (**B**) Lateral radiogram: neutral; (**C**) Mid-sagittal reconstruction CT; (**D**) Coronal reconstruction CT.

**Figure 4 brainsci-11-01241-f004:**
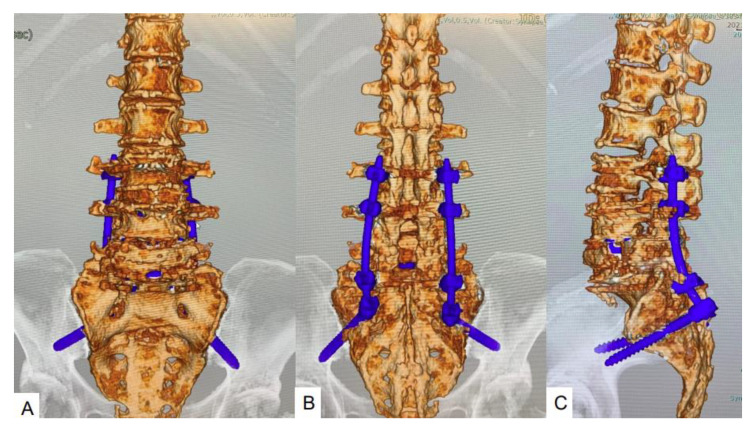
CT at final follow-up at one year. (**A**) Anteroposterior 3D CT; (**B**) Posteroanterior 3D CT; (**C**) Lateral 3D CT.

**Figure 5 brainsci-11-01241-f005:**
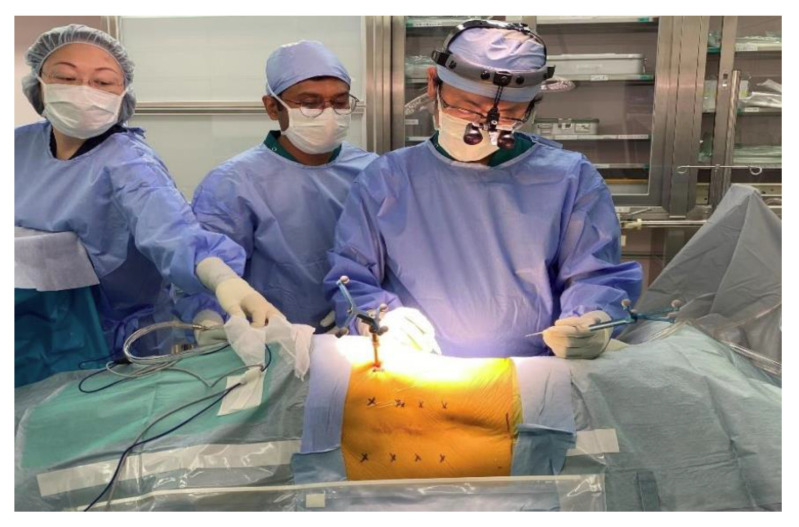
Position of the reference frame.

**Figure 6 brainsci-11-01241-f006:**
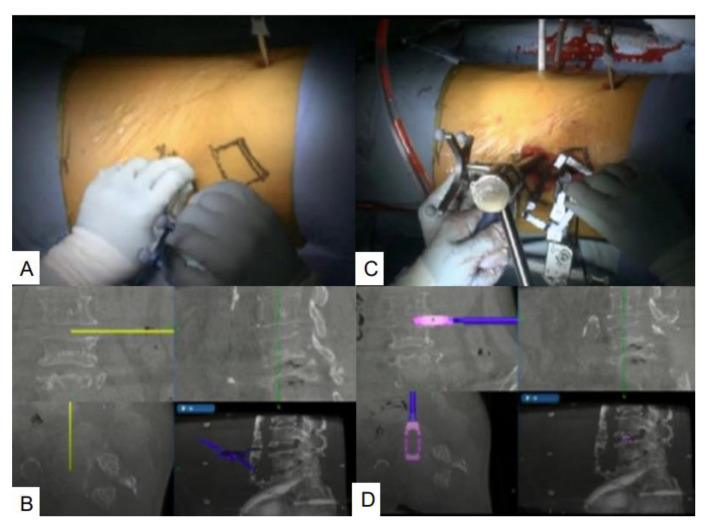
Navigated pointer and OLIF cage. (**A**,**B**) Skin incision with a navigated pointer; (**C**,**D**) Navigated OLIF cage.

**Figure 7 brainsci-11-01241-f007:**
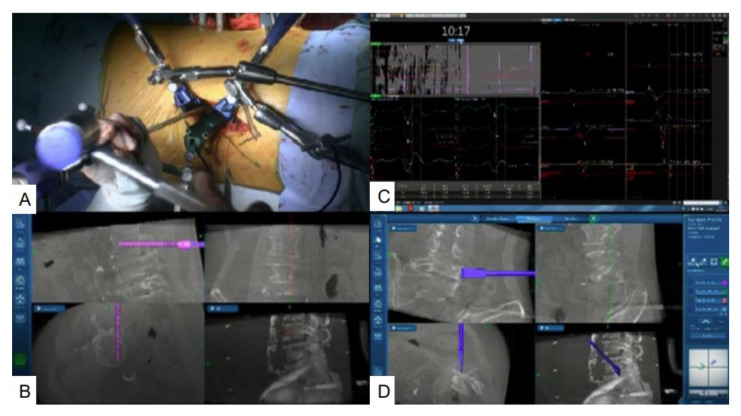
Simultaneous PPS and navigated osteotome. (**A**) Surgical field; (**B**) Simultaneous PPS; (**C**) Neuromonitoring; (**D**) Navigated osteotome.

**Figure 8 brainsci-11-01241-f008:**
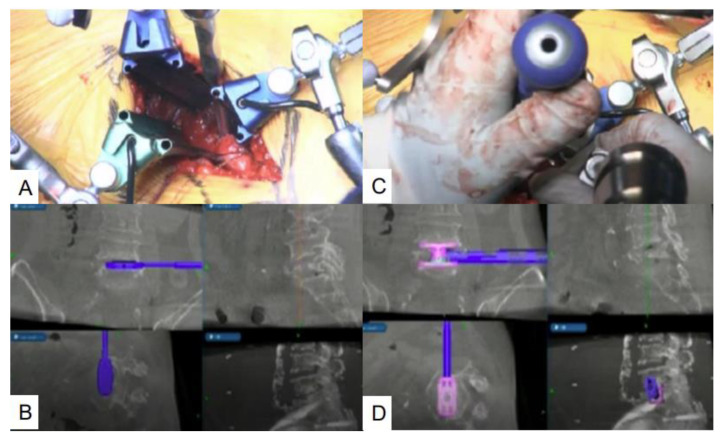
Navigated expandable cage, T2 Stratosphere^TM^ Expandable Corpectomy System. (**A**,**B**) Navigated trial; (**C**,**D**) Navigated expandable cage.

## Data Availability

Not applicable.

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
