# Peer review of "Minimally Invasive L5 Corpectomy with Navigated Expandable Vertebral Cage: A Technical Note"

_brainsci, 2021, doi:10.3390/brainsci11091241_

Round 1
Reviewer 1 Report
Well written and presented study regarding a Minimally Invasive L5 Corpectomy with Navigated Expandable Vertebral Cage.
Major points:
- for being classified as a technical note the manuscript is too long; anyhow the quality of the presentation, as well as the quality of the images is good. I would suggest to reduce the text and eliminate some images.
- At the end of the section discussion you should report limitation of the study, above all no conclusion should be drown considering that this is just a case report.
Minor points:
- some minor English grammar corrections.
- In the section discussion you could add some lines regarding the imaging of this disease, you could cite: Colosimo C et al. Imaging in degenerative spine pathology. Acta Neurochirurgica, Supplementum,
- In the section discussion you could add some lines regarding the utility and the differences of other procedures (vertebroplasty) compared to yours. In this case you could cite: Pedicelli A et al. Percutaneous vertebroplasty with a high-quality rotational angiographic unit. European Journal of Radiology. Volume 69, Issue 2, Pages 289 - 295February 2009
Author Response
To reviewer#1
Major points:
- For being classified as a technical note the manuscript is too long; anyhow the quality of the presentation, as well as the quality of the images is good. I would suggest to reduce the text and eliminate some images.
We appreciate your valuable comment.
We reduced the sentences and images as you suggested.
- At the end of the section discussion, you should report limitation of the study, above all no conclusion should be drown considering that this is just a case report.
Thank you for your important comment. We added the sentences as bellows;
There are several disadvantages of our new technique. First, the intraoperative CT-based spine surgery is not without the limitations of possible surgical errors involving misplacement of spinal implants due to inadvertent movement of the reference frame. Second, it needs higher establishment costs. Third, because of simultaneous technique, two surgeons are necessary for this technique.
Minor points:
- Some minor English grammar corrections.
Thank you for your comment.
We performed English grammar corrections.
- In the section discussion you could add some lines regarding the imaging of this disease, you could cite: Colosimo C et al. Imaging in degenerative spine pathology. Acta Neurochirurgica, Supplementum, 2011, (108), pp. 9–15
We appreciate your comment.
According to your advice, we added the sentences as follows;
For the diagnosis of OVF, MRI is clearly superior to both conventional radiography and CT and it should be preferred as first diagnostic examination [8].
- In the section discussion you could add some lines regarding the utility and the differences of other procedures (vertebroplasty) compared to yours. In this case you could cite: Pedicelli A et al. Percutaneous vertebroplasty with a high-quality rotational angiographic unit. European Journal of Radiology. Volume 69, Issue 2, Pages 289 - 295February 2009
Thank you for your advice. We added the sentence in the discussion section and reference [9] as follows;
Vertebroplasty and kyphoplasty for OVF is a standard technique. In severe cases like paralysis, MIS L5 corpectomy with a navigated expandable vertebral cage is a safe and effective technique.
Reviewer 2 Report
Authors present a case report on minimaly invasive navigated L5 corpectomy using exandable cage without fluoroscopy in lateral decubitus position. Satisfactory clinical outcome has been reported, with presentation of preoperative, intra- and postoperative imaging.
There are several issues which need to be clarified:
- The patient had osteoporosis and L5 Fracture; what was the ratio of performing only decompressive surgery - the standard in this case is percutaneous stabilization with cement augmentation, in cases where there are no deficits kyphoplasty - were these options discussed?
- The description of the second surgery is insufficient. According to postoperative imaging, the patient received not only expandable cage in the L5 following corpectomy but also a XLIF cage one level above; furthermore, the photos are not clear - was the approach only lateral (XLIF-approach) or did you perform OLIF approach (as you claim in the text) - were both cages implanted via the same route?
- Why did not you use cement augmentation of the screws for patient with osteoporosis?
- How did you perform the registration of the navigated cage? How did you perform the accuracy check?
- Please perform a literature review on cases of L5 corpectomy and the possiblities of graft - bone, non-expandable, expandable cage
- How long was your follow-up, are there any signs of fusion. What is your experience in subsidence with expandable cages?
Author Response
Reviewer #2
There are several issues which need to be clarified:
- The patient had osteoporosis and L5 Fracture; what was the ratio of performing only decompressive surgery - the standard in this case is percutaneous stabilization with cement augmentation, in cases where there are no deficits kyphoplasty - were these options discussed?
Thank you for your important comment.
As you mentioned, the first surgery might not be adequate for this patient. However, the patient was relatively old and unfortunately the augmented screws are not available in Japan.
- The description of the second surgery is insufficient. According to postoperative imaging, the patient received not only expandable cage in the L5 following corpectomy but also a XLIF cage one level above; furthermore, the photos are not clear - was the approach only lateral (XLIF-approach) or did you perform OLIF approach (as you claim in the text) - were both cages implanted via the same route?
We appreciate your valuable comment. Your opinion is perfect.
As you know, SAI screws are very strong anchors for pelvis (LIV). L 3-4 OLIF cage (same route) was inserted because the patient had severe osteoporosis. We thought 4 pedicle screws (UIV) are necessary to maintain alignment and prevent screws pullout. Again in Japan, augmented screws are not available.
- Why did not you use cement augmentation of the screws for patient with osteoporosis?
Thank you for your comment. In Japan, augmented screws are not available.
- How did you perform the registration of the navigated cage? How did you perform the accuracy check?
We appreciate your valuable comments.
The registration of the navigated cage is semi-automatic. The accuracy check is exactly the same of others navigated instruments. The surgeon should confirm the navigation accuracy by touching bony structures.
- Please perform a literature review on cases of L5 corpectomy and the possiblities of graft - bone, non-expandable, expandable cage
Thank you for your adequate suggestion. We added the sentences as follows;
Several options are available to reconstruct the collapsed vertebra, such as autogenous iliac crest bone graft, allograft, non-expandable cage [13] and expandable vertebral cage [4]. For severe osteoporotic patients, autogenous iliac bone graft is inadequate because of insufficient bone quality to maintain stability. The non-expandable cages are difficult to fit the resected vertebral space.
- How long was your follow-up, are there any signs of fusion. What is your experience in subsidence with expandable cages?
Thank you for your comment.
The follow-up period was just one year. Fortunately, there is no subsidence of an expandable cage of this cage at final follow-up. As you mentioned, we have several cases of cage subsidence, but not this case.
Reviewer 3 Report
Very well written manuscript, scientifically sound and really novel. Neverheless, I would to mention that, as 'The duration of the second surgery was 215 minutes, and estimated blood loss was 750 milliliters', this is in contradistinction with a MIS procedure.
Author Response
To reviewer#3
I would to mention that, as 'The duration of the second surgery was 215 minutes, and estimated blood loss was 750 milliliters', this is in contradistinction with a MIS procedure.
Thank you for your important comment.
We cannot argue with you about your comment. You are perfectly right. However, the procedure of L5 corpectomy is very difficult, as you know. According to the systemic review, the average blood loss was 3230ml in conventional posterior-anterior technique.
(D'Aquino D, Tarawneh AM, Hilis A, Palliyil N, Deogaonkar K, Quraishi NA. . Surgical approaches to L5 corpectomy: a systematic review. Eur Spine J. 2020, 29, 3074-3079. )

Round 2
Reviewer 1 Report
All the requested revisions have been done.
Reviewer 2 Report
The authors have adequatly referred to all questions, please add that the cement augmentation is an option which is not available in Japan.